# Assessing the Differential Methylation Analysis Quality for Microarray and NGS Platforms

**DOI:** 10.3390/ijms24108591

**Published:** 2023-05-11

**Authors:** Anna Budkina, Yulia A. Medvedeva, Alexey Stupnikov

**Affiliations:** 1Department of Biomedical Physics, Moscow Institute of Physics and Technology, 141701 Dolgoprudny, Russia; 2Federal State Institution «Federal Research Centre «Fundamentals of Biotechnology» of the Russian Academy of Sciences», 119071 Moscow, Russia

**Keywords:** differential methylation, microarrays, WGBS, RRBS, methylation signature, rank statistic, quality metric

## Abstract

Differential methylation (DM) is actively recruited in different types of fundamental and translational studies. Currently, microarray- and NGS-based approaches for methylation analysis are the most widely used with multiple statistical models designed to extract differential methylation signatures. The benchmarking of DM models is challenging due to the absence of gold standard data. In this study, we analyze an extensive number of publicly available NGS and microarray datasets with divergent and widely utilized statistical models and apply the recently suggested and validated rank-statistic-based approach Hobotnica to evaluate the quality of their results. Overall, microarray-based methods demonstrate more robust and convergent results, while NGS-based models are highly dissimilar. Tests on the simulated NGS data tend to overestimate the quality of the DM methods and therefore are recommended for use with caution. Evaluation of the top 10 DMC and top 100 DMC in addition to the not-subset signature also shows more stable results for microarray data. Summing up, given the observed heterogeneity in NGS methylation data, the evaluation of newly generated methylation signatures is a crucial step in DM analysis. The Hobotnica metric is coordinated with previously developed quality metrics and provides a robust, sensitive, and informative estimation of methods’ performance and DM signatures’ quality in the absence of gold standard data solving a long-existing problem in DM analysis.

## 1. Introduction

DNA methylation is an epigenetic mark that plays a significant role in many biological processes, such as regulating gene expression and chromatin remodeling (reviewed in [1]). Methylation contributes to normal mammalian development [2] as well as disease onset and progression, including cancer [3], neuro-generative [4] and metabolic disorders [5].

Differential methylation (DM) analysis is resulting in differentially methylated cytosines (DMCs), -probes (DMPs), or -regions (DMRs). Here, we refer to the list of DMCs or DMPs as a methylation signature (or simply a signature). A signature is supposed to accurately classify samples into two or more groups and capture the majority of the DNA methylation alterations associated with a disorder or other case state [6]. Many signatures for various types of diseases, such as cancer, metabolic disorders, and neuropsychiatric disorders have been recently identified [7,8,9,10,11]. DM signatures, alongside differential gene expression (DGE) signatures, play an important role in clinical and translational applications.

### 1.1. Evaluation Studies

Given the variety of available models for DM analysis, the choice of a particular method is a challenging task [12,13,14], making the methods benchmark and comparison critical for DM analysis. Numerous studies to evaluate DM analysis methods can be classified based on signature quality (1), methods similarity (2), and methods performance (3) (Table 1). Benchmark approaches based on quality metrics are the most numerous and can utilize simulated, real, or permuted data. Since no experimental “gold standard” is available for DM analysis, the most popular evaluation techniques involve simulated datasets. A “ground truth” list of DMCs or DMRs allowing for a larger number of available metrics to be estimated is predefined in such a design. However, simulated data are highly dependent on a chosen distribution and could lack the general characteristics of real bisulfite sequencing data, causing bias in the performance evaluation.

Benchmarks based on experimental data are limited to imprinted DMRs [15], gold-standard DMRs inferred from the results of RNA-seq and DNase-seq experiments [16] and methylation titration data [17]. While using real data, the assumptions and biases that are inevitable in the simulation approach can be avoided. Nevertheless, approaches with limited validation introduce biases of a different nature, as the “ground truth” signature may require additional verification, and only a part of the DMCs or DMRs present in real data could be included in such a signature, even with the help of Appendix A.

Benchmarks based on permutation of labels in compared groups [18] usually utilize an FPR metric. However, the FPR metric alone is not sufficient for a complete DM analysis methods assessment; for a comprehensive evaluation, it must be applied along with other criteria that estimate the number of true positives.

Concordance of the produced results can also serve as grounds for methods’ comparison. The similarity of the methods can be estimated by the percentage of overlapping DMCs between methods [13], the correlation between sets of *p*-values for each DMR [18], or Kendall’s coefficient of concordance for each pair of methods [14]. Estimating the results’ consistency can reveal the pairwise similarity of the evaluated methods but does not show the quality of the resulting signature of a particular method. The execution time and memory usage can be taken into consideration while comparing DM analysis methods [13,14,19]. Although comparing important user parameters, such methods do not evaluate the quality of the results obtained.

**Table 1 ijms-24-08591-t001:** Classification of DM methods evaluation studies.

Strategy	Criteria	Studies	Data
(1) Signature quality	Recall	[12,13,14,16,18]	Simulated
		[17]	Real
	True negative rate	[13,18]	Simulated
	False discovery rate	[12]	Simulated
	Precision	[14]	Simulated
		[17]	Real
	ROC AUC	[13,14,16,18,19]	Simulated
	False positive rate	[13,16,18]	Simulated
		[14]	Real
	Empirical distribution of *p*-values under the null	[13,16]	Simulated
	DMR overlapping fraction	[16]	Real
(2) Methods similarity	Ratio of overlapping DMCs between a pair of tests	[13]	Simulated
	Spearman correlation between *p*-values	[18]	Simulated, Real
	Kendall’s coefficient of concordance	[14]	Simulated
(3) Methods performance	Computation time	[13,14,19]	Simulated
	Computation memory	[14]	Simulated

### 1.2. Hobotnica Approach

Existing approaches for methods’ evaluation either lack quantitative metrics or require ‘ground truth’ data that limit their applicability to simulated or partly experimentally validated data and, therefore, significantly constrain their practical use. In addition, these approaches do not take into account dataset heterogeneity, while the metrics computed for a particular method may vary significantly across datasets. The label-permuting-based approach, although it can be applied for a particular real dataset, only allows for FPR computation, which can have limited sensitivity. Moreover, new methods are being developed constantly, making it critical to develop an approach that can evaluate the quality of the signature for the available dataset in an effective manner.

To address these problems, we applied the Hobotnica metric (H-score) [20] that we previously developed to assess the quality of molecular signatures obtained by the differential analysis of two or more groups of samples with different phenotypic characteristics and validated for DGE and DM signatures. In this way, H-scores of different DM signatures may be compared, allowing the direct evaluation of the models’ performance for a particular dataset by assessing the quality of phenotypes separation, delivered by a particular signature [21]. No metric has previously been developed that evaluates the DM signature’s quality in the context of a particular data set (e.g., inter- and intra-group samples distances) without a list of gold-standard DMC. Hobotnica provides a novel, gold-standard free approach for DM signature assessment.

### 1.3. Scopes and Objectives of the Study

In this study, with regard to DM models evaluation, we pursue the following tasks: ■To infer the influence of a data type (microarray vs. NGS derived) on the quality of DM signatures;■To evaluate the concordance of DM models within each data type;■To qualify the impact of signature’s subsetting;■To contrast the quality of DM signatures obtained on the simulated and real experimental NGS data;■To evaluate the relation and discordance between the H-score and existing quality metrics.

## 2. Results

### 2.1. Microarray Data

We conducted DM analysis for 16 contrasts on microarray datasets between case and control groups using limma, *T*-test, and dmpFinder with and without variance shrinkage. The sizes of the obtained DM signatures ranged from 1 to 78,359. For four datasets, none of the observed methods detected DM signatures. For one dataset, only a T-test returned a non-empty signature. Within the same dataset, signature lengths were rather similar, though they varied widely among datasets (Appendix A). Non-empty DM signatures obtained by all different methods had a large intersection of DMP (Figure 1A, Appendix A).

For the vast majority of non-empty DM signatures, the H-score value exceeded 0.7 (*p*-values < 0.05, Figure 1B,C, Appendix A). In two cases, no significant separation of groups was observed. For the GSE157341 dataset, only the *T*-test method returned a non-empty but non-significant signature of seven DM sites. For the GSE210301 dataset, two out of three comparisons received an H-score of 1 but with non-significant *p*-values.

To run Hobotnica for the signatures of reduced size and compare H-scores of the full signature and its smaller subsets, the top 100 DMP and top 10 DMP signatures were tested. For several datasets, H-scores of the truncated signatures were higher than H-scores of the full signature ((Figure 1D and Figure 2, Appendix A). The majority of the top 100 DMP and top 10 DMP signatures had a non-zero intersection (Appendix A).

The H-scores for different methods within the same dataset were rather close to one another (Figure 1D), which was to be expected since the signatures from different methods overlapped significantly. At the same time, they highly varied across different datasets (Appendix A). Full signatures as well as a subset of the top 100 and top 10 DMPs demonstrated no significant differences between the H-scores for the evaluated methods (Figure 2, Appendix A, *p*-value > 0.05 Friedman test, Appendix A).

### 2.2. Experimental NGS Data

For the three WGBS and three RRBS processed datasets, the DM signature lengths varied both between different methods and across different datasets (Appendix A). The methylSig method obtained non-empty DM signatures only for the two datasets. The methylKit with overdispersion correction returned only one DMC for the two datasets.

There were no DMCs detected by all the DM methods for all datasets, except GSE103886, which had a consensus signature of 12 DMCs shared by all methods (Figure 3B, Appendix A). Yet, for the methods based on beta-binomial distribution, a small fraction of DMC was shared (Figure 3A, Appendix A). Given this, it is no surprise that, unlike H-scores for microarray data, for NGS data, H-scores varied dramatically (Appendix A, Appendix A). For dataset GSE150592, H-scores ranged from 0.54 to 0.98 (Figure 3E) with all of them except for HMM-DM being significant (*p*-value < 0.05, Appendix A). The highest H-score was obtained by DSS without smoothing (Figure 3C,D). In contrast, for datasets GSE138598 and GSE103886, all signatures received H-scores nearly equal to 1 with a small variance across methods.

In contrast to the H-scores obtained for microarray data, H-scores for NGS data did not show consistent improvement for a shorter signature of the top 100 DMC and top 10 DMC (Figure 3E and Figure 4, Appendix A). The top 100 and top 10 DMC signatures both for methods based on the beta-binomial distribution (Appendix A), and all methods (Appendix A) had zero intersection for the majority of datasets. For several methods (methylKit, DSS, and BSmooth), the top 10 DMC signatures returned a zero H-score. The DM methods’ results were significantly different based on the resulting H-scores (*p*-value < 0.05, Friedman test) for all WGBS and RRBS datasets (all methods except BSmooth were tested) (Appendix A, Figure 4).

### 2.3. Simulated NGS Data

For all simulated datasets, the length of the DM signatures increased in concordance with the specified methylation difference (Appendix A). methylKit produced much shorter signatures than the other methods, while DSS with smoothing and RADMeth produced the largest signatures for both groups of datasets.

In contrast to the results for experimental NGS data, DM signatures for simulated data have large intersections not only for beta-binomial methods (Appendix A) but for all methods as well (Appendix A). The signatures of the HMM-DM method overlap almost completely with signatures from beta-binomial methods (Appendix A). The methylKit signatures have minimal overlap with other methods. In most cases, there is no intersection between the shortened top 100 DMC or top 10 DMC signatures (Appendix A).

For most methods, recall, precision, and accuracy improved with the increase in methylation difference (Figure 5A–C, Appendix A). The DSS with smoothing and RADMeth methods had the highest recall values relative to other methods. Except for methylKit, all methods’ average precision values were higher than 0.75, with a greater variation in precision for lower methylation differences. DSS with smoothing and RADMeth had the highest precision, which slightly increased with an increase in the methylation difference. Accuracy patterns for different methods were similar to precision.

The H-scores for all methods except methylKit are close to 1, even for a methylation difference of 0.1, and slightly increase with an increase in the difference in methylation (Figure 5E), opposite of the results obtained for the experimental NGS data. Both precision and recall metrics varied significantly for different methods, and the value of each metric increased with the difference in methylation. Differently, the H-scores did not significantly alter depending on the methylation difference values and were close to 1, with only methylKit signatures being characterized by both low precision and recall and low H-scores close to 0.5.

Since Hobotnica by design characterizes the ability of a signature to separate groups of samples, we explored the connection of H-score patterns with regard to the precision metric (Figure 5F, Appendix A). The H-score for such signatures correlates with the precision, and H-score growth is more rapid for greater methylation difference levels (Appendix A).

The label permutation-based FPR was quite low and comparable across most methods, with RADMeth having the highest rate, and remained consistent regardless of the changes in methylation difference for most methods except RADMeth and HMM-DM (Figure 5E, Appendix A). Most H-scores for label permutation were equal to zero due to the short signature length or close to 0.5 (Appendix A).

In a few cases, signatures consisting of false positives received high H-scores with a significant separation. Even though there was no visible difference between the absolute values of intragroup and intergroup distances (Appendix A), ranking the distances made the variation noticeable (Appendix A).

## 3. Discussion

Selecting a suitable method for DM analysis is challenging due to the large and increasing number of available models. Model assessment can be an issue due to the lack of gold standard data. Existing evaluation approaches are indirect or use simulated data that constrain their efficiency and applicability. In this study, we present an evaluation strategy for microarray and NGS datasets based on the previously developed Hobotnica approach. We demonstrate that the DM signatures based on the microarray data are of good quality and highly convergent: the signatures produced by the methods in all datasets have comparable lengths and a significant overlap, which is reflected in highly significant and very similar H-scores. On the other hand, DM analysis based on NGS data is inconsistent across methods. Different methods return signatures of significantly different lengths and content for the same dataset, which is reflected in variable and often non-significant H-scores.

Results obtained for the simulated NGS data differ significantly from those obtained for experimental NGS data. The content of DM signatures for simulated data shows higher convergence, and corresponding H-scores have higher values with less variability. This illustrates that existing simulation tools do not fully reflect the complexity of real data. Therefore, the results of the evaluation based on simulated data should be considered with caution. Thus, having in mind that simulated NGS data cannot be considered a “gold standard”, the direct strategy of the evaluation, such as the Hobotnica approach, is critical for the DM quality estimation.

Although hundreds or thousands of statistically significant DMC can be detected, evaluating the effective length of a signature can assist with most practical applications. However, it is often not clear whether a subset signature of top DMC incorporates ample relevant information. Despite the fact that, in some cases, extra-short signatures of 10 DMC were substantial for data stratification (especially for microarray data), in other cases, short signatures delivered lower H-scores and worse data separation (or even no separation for the shortest subsets, which resulted in a zero H-score in some cases), compared to full-length signatures (notably for NGS data). Thus, the length of a subset signature should be chosen not only considering pragmatic reasons (necessary for the interpretation or sufficient for diagnostics) but needs to be explicitly tested in regard to its quality.

Detecting and assessing an approach’s limitations is as important as its design or validation. In this work, we limited the Hobotnica applicability to DMC only, while the DMR format for methylation signatures is often more widely used and more easily interpreted. The DMR format, however, poses severe problems for signatures comparison since setting a formal criterion, whether two intersecting yet differing regions should be treated as the same entity or not, is a challenging task. In addition, each region can be characterized not only by the mean level of methylation but also by the length of the region, as well as the density of sites in it.

In our study, we detected other scenarios that may constrain Hobotnica under specific conditions. The Hobotnica approach may lose sensitivity when applied for extra-short signatures. During pairwise distances computation, two samples may not have shared DMC, covered in both samples. The Euclidean distance between samples in such a scenario cannot be calculated. This can be addressed by increasing the tested signature’s length. In other cases, Hobotnica may be oversensitive and deliver high scores for false-positive signatures with the existing, albeit negligible, stratification effect. Practically, these cases, although found to be quite rare, can easily be distinguished by assessing not-ranked sample distances, which is a part of the Hobotnica analysis workflow. For the rest of the cases, Hobotnica was shown to provide a meaningful, sensitive, and robust evaluation of DM signatures for all platforms.

Our study provides the following practical recommendations for DM analysis. Micro-array-based approaches due to high convergence and performance should be favored when the study design allows, for specific applications, demanding increased robustness and recruiting known methylation sites as most transnational and clinical applications. NGS-based datasets, due to the high variability of the results obtained by different methods, should be processed with several different DM methods, and the resulting signatures should be validated either with limited experimental data or with Hobotnica, which allows for valid quality estimation of DM analysis performance for a newly generated dataset in the absence of gold standard data. The evaluation of newly developed methods for DM analysis should not be performed only on simulated data due to a significant bias in the results. This bias should be at least partially compensated for with tests on ’ground truth’ data, no matter how limited, as well as experimental datasets for evaluating which Hobotnica provides means.

## 4. Materials and Methods

### 4.1. Microarray Datasets

#### 4.1.1. Data

Microarray DM methods were evaluated on 14 datasets (450k Human Methylation Array and EPIC datasets, Table 2). Each dataset contains a case group representing a disease or a perturbation exposure and a control group. A dataset GSE210301 contains three case and one control group, resulting in three pairwise comparisons. Across tested datasets, the number of samples per group varied from 4 to 345 (Appendix A).

#### 4.1.2. Microarray Data Preprocessing

All datasets were preprocessed in a uniform way (Appendix A). The methylation data were loaded from IDAT intensity files. Each dataset was processed using the ChAMP package [22]. The intensity data were filtered based on detection *p*-values and bead count with default thresholds and imputation disabled. Next, the data were normalized using beta-mixture quantile (BMIQ) normalization. The result β-value matrix was used as an input for differential methylation analysis methods and Hobotnica calculation.

**Table 2 ijms-24-08591-t002:** Datasets for differential methylation analysis methods evaluation.

GEO Code	Reference	Cell Type	Description	Protocol	# Control	# Case
GSE210301	[23]	IMR90 cells	Cortisol exposure compared to vehicle	EPIC	6	6
GSE210301	[23]	IMR90 cells	Relacorilant exposure compared to vehicle	EPIC	6	6
GSE210301	[23]	IMR90 cells	Cortisol and Relacorilant exposure compared to vehicle	EPIC	6	6
GSE175458	[24]	lung tissue	Idiopathic pulmonary fibrosis compared to non-diseased control	EPIC	202	345
GSE175399	[25]	adipose/connective tissue	Thyroid-associated ophthalmopathy compared to control	EPIC	4	4
GSE210484	[26]	cultured primary fibroblast	Arboleda–Tham syndrome compared to control	EPIC	13	12
GSE196007	[27]	classical monocyte	Systemic sclerosis compared to control	EPIC	12	12
GSE156994	[28]	whole blood	Sporadic Creutzfeldt–Jakob disease compared to control	450K	105	114
GSE157341	[29]	liver tissue	Control compared to hepatocellular carcinoma	450K	35	228
GSE101764	[30]	mucosa tissue	Adjacent non-tumor compared to colorectal cancer	450K	149	112
GSE85845	[31]	lung tissue	Adjacent non-tumor compared to lung adenocarcinoma	450K	8	8
GSE156669	[32]	normal buccal mucosa	Oral submucous fibrosis compared to control	450K	5	7
GSE178218	[33]	LSCC and adjacent tissue	Laryngeal squamous cell carcinoma compared to control	450K	11	20
GSE178216	[33]	OSCC and adjacent tissue	Oral squamous cell carcinoma compared to control	450K	7	15
GSE178212	[33]	ESCC and adjacent tissue	Esophageal squamous cell carcinoma compared to control	450K	16	24
GSE157272	[34]	prostate tissue	Agressive prostate cancer compared to benign prostate tissue	450K	10	8
GSE149608	[35]	esophagus and ESCC tissue	Normal and esophageal squamous cell carcinoma tumor samples	WGBS	10	10
GSE138598	[36]	spermatozoa	Type 2 diabetes mellitus compared to control	WGBS	9	8
GSE119980	[37]	human cortex brodmann area 9	Rett syndrome compared to control	WGBS	6	6
GSE150592	[38]	primary dermal fibroblasts	Systemic sclerosis compared to control	RRBS	15	15
GSE148060	[39]	sural nerve	Comparing patients with the highest HbA1c levels to those with the lowest (control)	RRBS	32	21
GSE103886	[40]	liver tissue	STAT5a//STAT5b knockout mice compared to control	RRBS	11	12

#### 4.1.3. Differential Methylation Analysis

DM analysis was performed using four methods: limma with M-values input, two-sided Welch *T*-test using the β-values, and dmpFinder function from minfi package with and without variance shrinkage option enabled using the β-values. *T*-test and limma functions were executed with the RnBeads package [41]. For limma and *T*-test, a list of differentially methylated CpG sites was obtained using the FDR [42] threshold of 0.05. For dmpFinder, a q-value [43] threshold of 0.05 was applied. Only CpG sites with a mean methylation difference between groups ≥ 0.15 were used as the chosen signature for each method.

### 4.2. NGS Datasets

#### 4.2.1. Real Data

Six NGS datasets (WGBS and RRBS) were used for differential methylation analysis (Table 2). The number of samples per group ranged from 6 to 32 (Appendix A).

#### 4.2.2. NGS Data Preprocessing

Both WGBS and RRBS datasets were preprocessed in a similar manner, except for a deduplication step additionally performed for WGBS data only (Appendix A). Reads from each sample were trimmed using Trim Galore (version 0.6.6) [44] to filter out low-quality reads and cut adapters. Two base pairs were removed from the 3’ end of read 1 and the 5’ end of read 2 of the adapter-trimmed sequences for RRBS data.

The reads were mapped to human reference genome hg38 for GSE149608, GSE148060, GSE150592, GSE138598, and GSE119980 datasets, and to mm39 for GSE103886 dataset using Bismark (version 0.22.3) [45]. WGBS dataset reads were deduplicated using Bismark. For each CpG site, the methylation level was obtained using the Bismark methylation extractor. Only the CpG sites with coverage greater than five reads were held for further analysis. Sites with no coverage in at least two samples of each group were discarded.

#### 4.2.3. Differential Methylation Analysis

Six software packages were applied to each dataset for DMC identification between two groups: methylKit [46], BSmooth [47], DSS [48,49,50], MethylSig [51], RADMeth [52], and HMM-DM [53]. DSS was used with and without prior methylation level smoothing. methylKit was applied with and without overdispersion correction. Other methods were used with the default parameters.

DMCs with a default *p*-value adjustment lower than 0.05 were included in a signature for the MethylSig, methylKit, DSS, and RADMeth methods. Signatures were sorted based on the adjusted *p*-value.

The BSmooth smoothing procedure was applied to methylation level values before low-coverage filtering. After smoothing, only the sites with a methylation ratio defined for all samples were selected for further DMC identification. The tested CpG sites were sorted based on the corresponding absolute t-statistics value. Sites that had t-statistics between the 5% and 95% quantiles were included in the resulting signature. BSmooth was not applied to the RRBS datasets.

HMM-DM was applied to each chromosome separately with default parameters. DMCs in hypomethylated and hypermethylated states with posterior probabilities greater than 0.95 were added to the resulting signature. The posterior probability value was used to sort the sites in the final HMM-DM signature.

DMCs with absolute methylation differences lower than 0.15 were excluded from the signature for all the methods.

#### 4.2.4. Simulated RRBS Data

Two groups of ten RRBS paired-end simulated datasets were generated using the RRBSSim simulator [54]. The first group of datasets was prepared with default settings. For each dataset, 32 samples (16 case and 16 control samples) were generated. Chr22 (hg38) was used for simulations. For the second group, the mean CpG methylation level, sequencing depth, read length, and probability matrix of quality value counts were taken from the real dataset (GSE103886). For every individual simulated sample, all these parameters, except for the read length, were derived from a specific sample of the real dataset. For each dataset, 11 case and 12 control samples were simulated as in the real dataset. Chr19 (mm39) was used for simulation.

The resulting FASTQ files were processed in the same way as real NGS datasets. In a group of case samples, the difference in the CpG methylation levels was introduced similarly to that described in [14]. For each dataset, reads from 300 randomly selected CpG islands (CGI) provided by UCSC Genome Browser [55] were simulated with methylation differences of 0.1, 0.15, 0.2, and 0.3. Of all selected CGI, 150 regions were set as hypomethylated, and 150 regions were set as hypermethylated.

In each experiment, all DM methods applicable to RRBS data were tested: MethylSig, DSS with and without smoothing, methylKit with and without overdispersion correction, RADMeth, and HMM-DM. Precision, recall, accuracy, and H-score metrics were calculated for the obtained signatures. To calculate the false positive rates, we permuted the sample labels once for each simulation. All obtained DMCs were considered false positives, and the rest of the covered CpG were considered true negatives. To investigate the relationship between the H-score patterns and the precision metric, the dependence of the H-score on the precision value was calculated for a set of synthetic signatures of the same length as a ground truth signature with a different proportion of the true positive DMCs.

### 4.3. Hobotnica

Hobotnica [20] evaluates signatures based on the distance values between samples, which is inferred as the distance between vectors from the molecular signature subset of molecular features (CpG site positions). For differential methylation analysis, each vector contains methylation level values. Following that, the distances between samples are ranked, as ranking makes the metric more robust to the distance selection and helps to mitigate the impacts of outliers. The statistical significance of the H-score can be assessed by calculating an empirical permutation *p*-value from a distribution of H-scores of random signatures with the same length.

The H-score was calculated for each signature and its smaller subset (top 100 and 10 DMCs from the signature sorted by adjusted *p*-value). Hobotnica was applied to beta values (microarrays) and methylation ratio values (NGS) with Euclidean distance. The H-score was set to zero if a distance matrix contained at least one NA entry or a method did not return any DMCs. H-scores were defined only for signatures of length greater than or equal to two sites. Whether an empty signature should be dismissed from the evaluation (since no false result was returned by the method) or needs to be set to the minimum, i.e., 0 (given no possible stratification to groups can be performed for samples), is a precarious question. We evaluated both scenarios and assessed the results separately.

To calculate the *p*-value for each signature, we sampled 5000 random signatures of the same length with replacement from the corresponding dataset and calculated the H-score for each of them. H-scores equal to zero were not included in the final distribution. If the number of sampled H-scores was less than 5000, the resulting *p*-value was not defined. The *p*-value was computed by incorporating a pseudo-count.

### 4.4. Statistical Inference

H-score distributions of the observed DM methods were compared to detect differences between the methods. First, the H-scores distributions from microarray and NGS results were tested for normality using the Shapiro–Wilk normality test. The Friedman test [56] was applied because the H-scores distributions did not meet the ANOVA test assumption of the independent observations and the normality assumption. H-scores equal to zero were included in the test. The Friedman test statistics were calculated for the WGBS and RRBS datasets values separately and for all datasets, not including the BSmooth method, as it cannot be applied to RRBS data. The benchmark is available at https://github.com/lab-medvedeva/Hobotnica-DiffMeth-comparison (accessed on 28 April 2023).

## 5. Conclusions

In this study, we performed a benchmark for DM methods on multiple NGS and microarray datasets based on the resulting DM signature’s quality. We applied the rank statistic approach Hobotnica to assess models’ performance in the absence of gold standard data. The observed heterogeneity of signatures’ quality across experimental platforms, DM models, and biological datasets confirmed the necessity for the signatures’ quality assessment in newly conducted analyses. Hobotnica provides provide robust, sensitive, and informative estimation for DM signature quality, solving a long-existing problem in DM analysis.

## Figures and Tables

**Figure 1 ijms-24-08591-f001:**
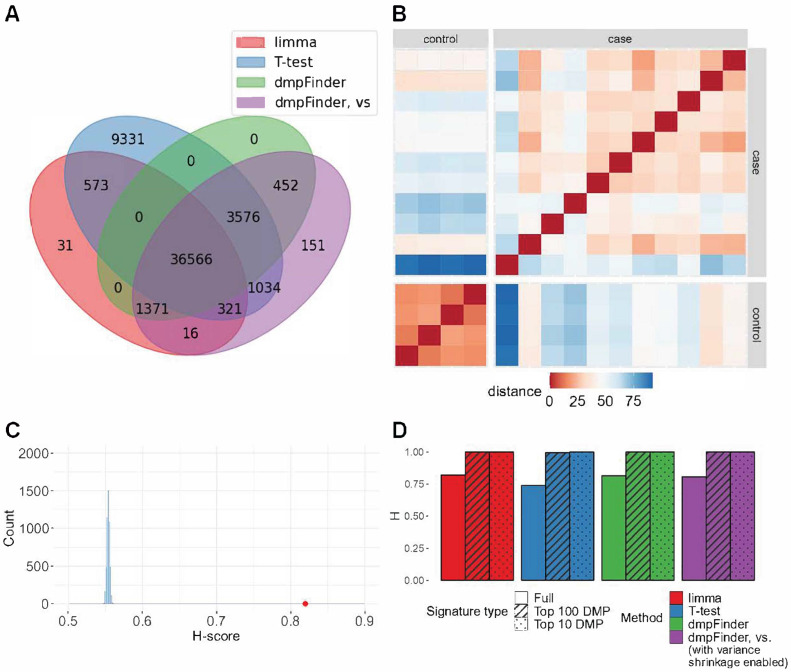
Processing of GSE178216 dataset (microarray): oral squamous cell carcinoma (15 samples) compared to control (7 samples). (**A**) Intersection of signatures return by limma, *T*-test, dmpFinder and dmpFinder vs. (with variance shrinkage enabled). (**B**) The distances between the vectors of methylation levels of limma signature sites. (**C**) H-score for the full signature returned by limma method = 0.8197 (red), H-score distribution of random signatures of the same length as the result (blue). (**D**) H-scores for the full, top 100 DMP and top 10 DMP signatures.

**Figure 2 ijms-24-08591-f002:**
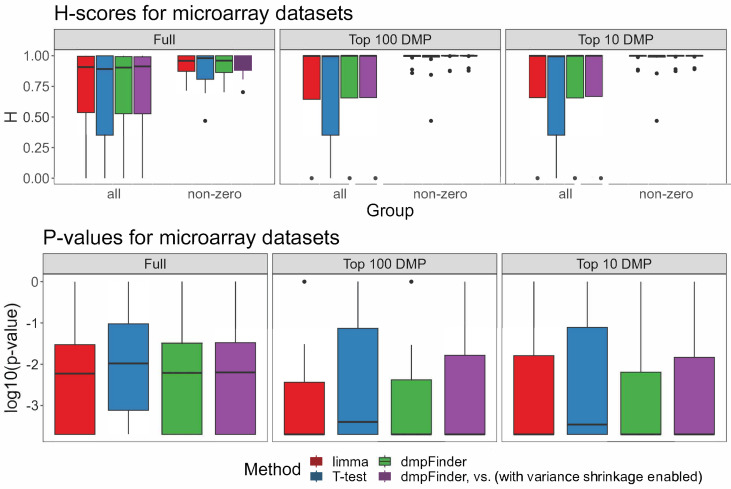
H-scores and *p*-values for the signatures of each method for microarray data, full, top 100 and top 10 DMP signatures (including or excluding H-scores = 0).

**Figure 3 ijms-24-08591-f003:**
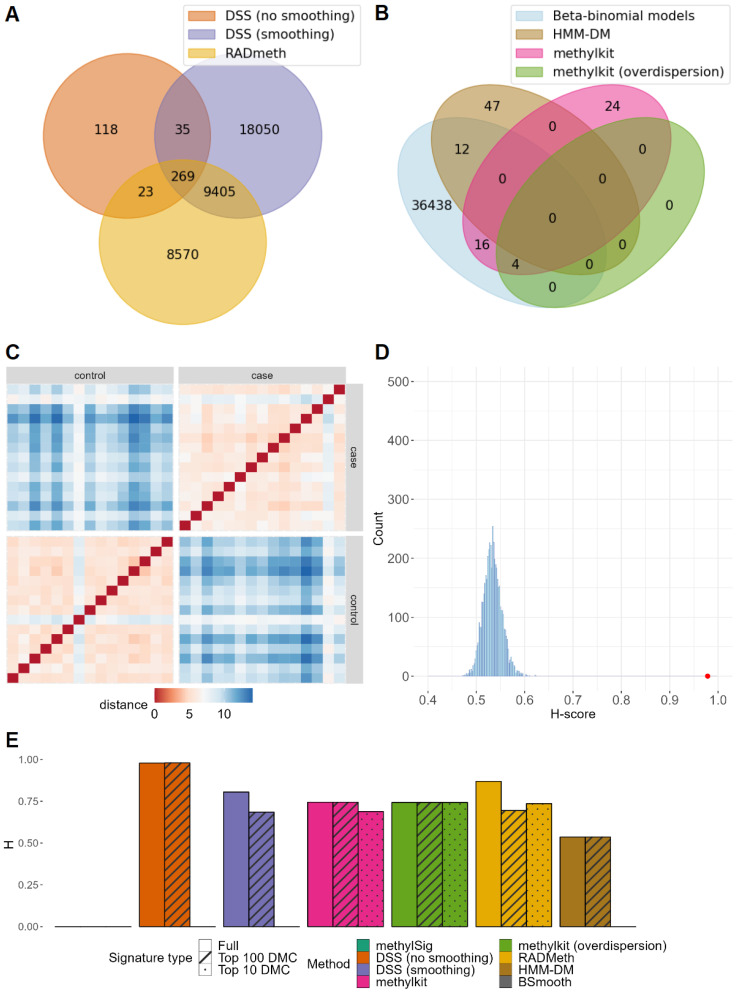
Processing of GSE150592 dataset (NGS): primary dermal fibroblasts from patients with systemic sclerosis (15 samples) compared to control (15 samples). (**A**) Intersection of signatures returned by beta-binomial model-based methods. (**B**) Intersection of the union of DMCs returned by beta-binomial model-based methods (blue) and the signatures from the rest of the methods. (**C**) Distances between vectors of methylation levels of DSS without smoothing signature sites. (**D**) H-score for the full signature returned by DSS without smoothing method = 0.9785 (red), H-score distribution of random signatures of the same length as the result (blue). (**E**) H-scores for the full, top 100 DMC and top 10 DMC signatures.

**Figure 4 ijms-24-08591-f004:**
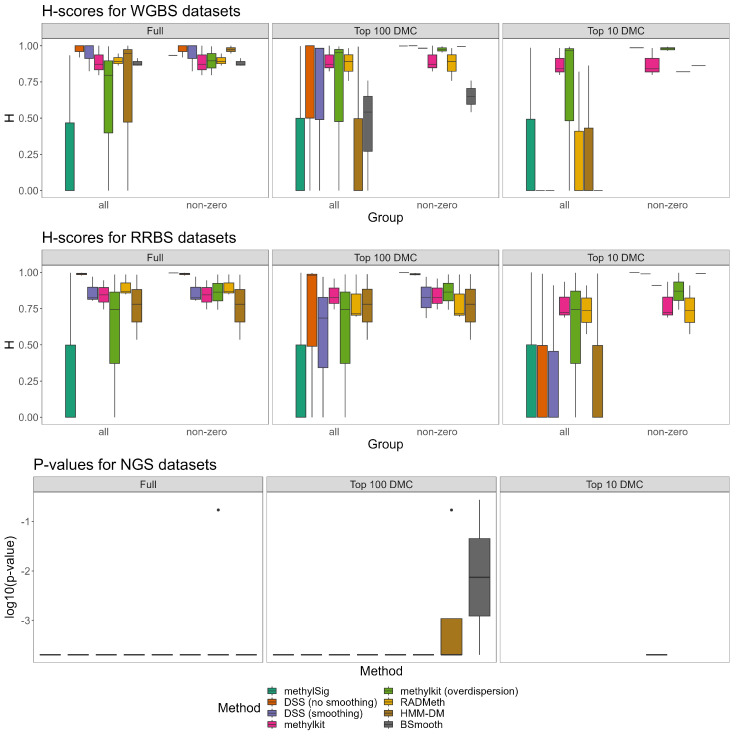
Hobotnica metric values and *p*-values for each method for NGS data, full, top 100 and top 10 DMC signatures (including or excluding H-scores = 0).

**Figure 5 ijms-24-08591-f005:**
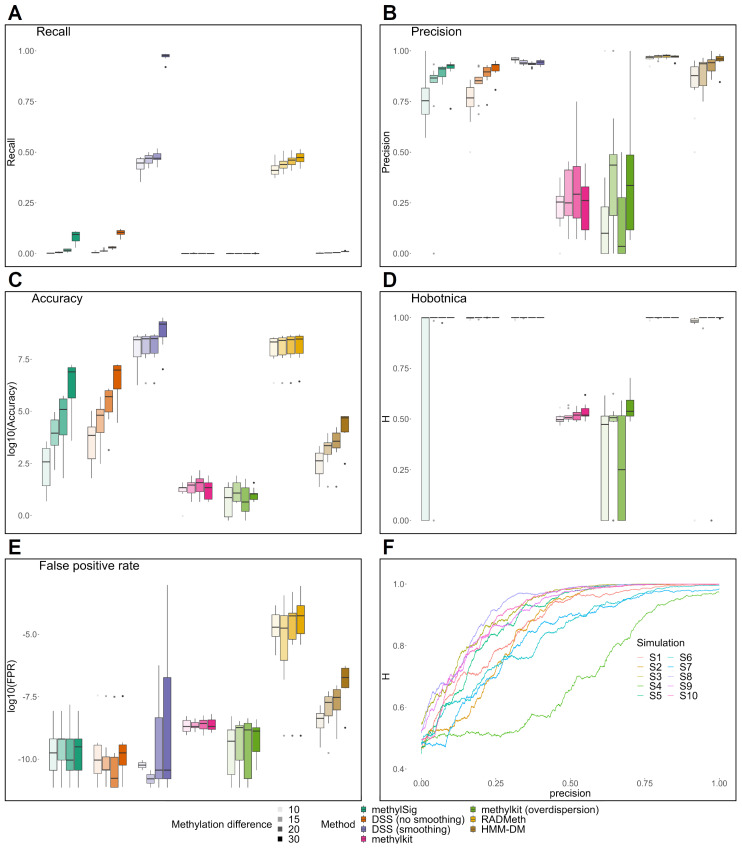
Results for DM analysis on simulated data (group of datasets simulated with parameters derived from real data: (**A**) recall, (**B**) precision, (**C**) accuracy for signatures produced from datasets simulated with methylation differences. (**D**) Hobotnica for signatures produced from datasets simulated with methylation differences. (**E**) False positive rate for signatures obtained after permuting sample labels for each simulated dataset. (**F**) Dependence of the H-score on the proportion of true positive DMC included in the signature. A signature had the same length as a ground truth signature for each simulation with an average difference in methylation levels of 0.1.

## Data Availability

Publicly available datasets were analyzed in this study. These data can be found here: GSE210301 https://www.ncbi.nlm.nih.gov/geo/query/acc.cgi?acc=GSE210301 (accessed on 15 February 2023), GSE175458 https://www.ncbi.nlm.nih.gov/geo/query/acc.cgi?acc=GSE175458 (accessed on 15 February 2023), GSE175399 https://www.ncbi.nlm.nih.gov/geo/query/acc.cgi?acc=GSE175399 (accessed on 15 February 2023), GSE210484 https://www.ncbi.nlm.nih.gov/geo/query/acc.cgi?acc=GSE210484 (accessed on 15 February 2023), GSE196007 https://www.ncbi.nlm.nih.gov/geo/query/acc.cgi?acc=GSE196007 (accessed on 15 February 2023), GSE156994 https://www.ncbi.nlm.nih.gov/geo/query/acc.cgi?acc=GSE156994 (accessed on 15 February 2023), GSE157341 https://www.ncbi.nlm.nih.gov/geo/query/acc.cgi?acc=GSE157341 (accessed on 15 February 2023), GSE101764 https://www.ncbi.nlm.nih.gov/geo/query/acc.cgi?acc=GSE101764 (accessed on 15 February 2023), GSE85845 https://www.ncbi.nlm.nih.gov/geo/query/acc.cgi?acc=GSE85845 (accessed on 15 February 2023), GSE156669 https://www.ncbi.nlm.nih.gov/geo/query/acc.cgi?acc=GSE156669 (accessed on 15 February 2023), GSE178218 https://www.ncbi.nlm.nih.gov/geo/query/acc.cgi?acc=GSE178218 (accessed on 15 February 2023), GSE178216 https://www.ncbi.nlm.nih.gov/geo/query/acc.cgi?acc=GSE178216 (accessed on 15 February 2023), GSE178212 https://www.ncbi.nlm.nih.gov/geo/query/acc.cgi?acc=GSE178212 (accessed on 15 February 2023), GSE157272 https://www.ncbi.nlm.nih.gov/geo/query/acc.cgi?acc=GSE157272 (accessed on 15 February 2023), GSE149608 https://www.ncbi.nlm.nih.gov/geo/query/acc.cgi?acc=GSE149608 (accessed on 1 May 2022), GSE138598 https://www.ncbi.nlm.nih.gov/geo/query/acc.cgi?acc=GSE138598 (accessed on 1 May 2022), GSE119980 https://www.ncbi.nlm.nih.gov/geo/query/acc.cgi?acc=GSE119980 (accessed on 1 May 2022), GSE150592 https://www.ncbi.nlm.nih.gov/geo/query/acc.cgi?acc=GSE150592 (accessed on 1 May 2022), GSE148060 https://www.ncbi.nlm.nih.gov/geo/query/acc.cgi?acc=GSE148060 (accessed on 1 May 2022), GSE103886 https://www.ncbi.nlm.nih.gov/geo/query/acc.cgi?acc=GSE103886 (accessed on 1 May 2022).

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
