# Peer review of "Assessing the Differential Methylation Analysis Quality for Microarray and NGS Platforms"

_ijms, 2023, doi:10.3390/ijms24108591_

Round 1

Reviewer 1 Report

Budkina et al. apply Hobotnica to evaluate the quality of methylation signatures on NGS and microarray dataset and show that microarray based methods are more robust. The manuscript is insightful and needs minor points to be addressed before publication:

1)      In Figure 2, H-scores for top 10 DMP signatures are not shown as mentioned in the text. True/ false should be added to every bar. Include p-values.

2)      In Figure 3E, H-scores for top 10 DMP signatures are not shown for some methods. Explain the discrepancy.

3)      In Figure 4, true/false are not indicated in the plot.

4)      Figure 5 needs to be better resolution. It is hard to read the methylation difference within the plots.

Author Response

Point 1: In Figure 2, H-scores for top 10 DMP signatures are not shown as mentioned in the text. True/ false should be added to every bar. Include p-values.

Response 1: We thank the reviewer for their suggestions. Fig. 2 in the manuscript was amended to include the requested additional information. 

Point 2: In Figure 3E, H-scores for top 10 DMP signatures are not shown for some methods. Explain the discrepancy.

Response 2: An explicit note for the observed pattern is included in the manuscript.

line 143: 

“For several methods (methylkit, DSS, BSmooth) top 10 DMC signatures returned zero H-score.”

line 212:

“Despite the fact, that in some cases extra short signatures of 10 DMC were substantial for data stratification (especially for microarray data), in other cases short signatures delivered lower H-scores and worse data separation (or even no separation for the shortest subsets, which resulted in zero H-score in some cases) compared to full-length signatures (notably for NGS data).”

Point 3: In Figure 4, true/false are not indicated in the plot.

Response 3: Fig. 4 in the manuscript was amended to include the requested additional information. 

Point 4: Figure 5 needs to be better resolution. It is hard to read the methylation difference within the plots.

Response 4: Fig. 5 in the manuscript was amended to include the requested additional information. 

Reviewer 2 Report

Thank you for giving me the opportunity to review your manuscript entitled "Assessing the Differential Methylation Analysis Quality for Microarray and NGS Platforms". However, after careful consideration, I regret to inform you that your manuscript does not meet our standards for publication.

I have several concerns regarding your study. Firstly, the manuscript lacks clarity in terms of its objectives and scope of research. Secondly, the manuscript appears to focus solely on evaluating statistical models for differential methylation analysis without providing any novel insights or findings. Thirdly, the abstract fails to provide sufficient details on the methodology used, which makes it difficult for readers to understand the study design.

Moreover, I have noted that your manuscript does not offer any practical implications or applications of the proposed rank-statistic-based approach, named Hobotnica. Additionally, the limitations and potential biases of the study are not adequately addressed.

In light of these concerns, I regret to inform you that your manuscript is not suitable for publication in our journal. I appreciate your interest in our publication and encourage you to revise your manuscript, taking into consideration the above-mentioned points, and submit it again for further review.

The manuscript contains several grammatical errors, awkward sentence structures, and unclear phrases, which make it difficult for readers to understand the content.

Author Response

Point 1: Firstly, the manuscript lacks clarity in terms of its objectives and scope of research.

Response 1: We thank the reviewer for pointing out this issue. To articulate the objectives of the study more clearly, we have rewritten the paragraph in more explicit terms and moved to the ‘Scopes and objectives of the study’ subsection.

line 86:

“1.3. Scopes and objectives of the study

In this study, in regard to DM models evaluation, we pursue the following tasks:

  • to infer the influence of a data type (microarray vs NGS derived) on the quality of DM signatures;
  • to evaluate the concordance of DM models within each data type;
  • to qualify the impact of signature’s subsetting; 
  • to contrast the quality of DM signatures obtained on the simulated and real  experimental NGS data;
  • to evaluate the relation and discordance between the H-score and existing quality metrics.”

Point 2: Secondly, the manuscript appears to focus solely on evaluating statistical models for differential methylation analysis without providing any novel insights or findings.

Response 2: To articulate the novelty of the approach and our findings, we have added shortened explanation to the “Abstract” and “Hobotnica approach” sections

line 14:

Summing up, given the observed heterogeneity in NGS methylation data, the evaluation of newly generated methylation signatures is a crucial step in DM analysis.

line 82:

“No metric has previously been developed that evaluates the DM signature's quality in the context of a particular data set (e.g. inter- and intra-group samples distances) without a list of gold-standard DMC. Hobotnica provides a novel, gold-standard free approach for DM signature assessment.”

Point 3: Thirdly, the abstract fails to provide sufficient details on the methodology used, which makes it difficult for readers to understand the study design.

Response 3: To specify key details of the recruited in the study  methodology we added a paragraph in the “Abstract” section

line 5:

The benchmarking of the models is challenging due to the absence of gold standard data. In this study we analyze an extensive number of publicly available NGS and microarray datasets with divergent and widely utilized statistical models and apply a recently suggested and validated rank-statistic-based approach, Hobotnica, to evaluate the quality of their results.”

Point 4: Moreover, I have noted that your manuscript does not offer any practical implications or applications of the proposed rank-statistic-based approach, named Hobotnica. 

Response 4: To correct this issue, we have added a paragraph, explicitly discussing the practical implication of the designed approach and the observed findings in the “Discussion” section

line 238:

Our study provides the following practical recommendations for DM analysis. Microarray-based approaches due to high convergence and performance should be favored when the study design allows, for specific applications, demanding increased robustness and recruiting known methylation sites, as most transnational and clinical applications. NGS-based datasets due to the high variability of the results obtained by different methods should be processed with several different DM methods and the resulting signatures should be validated either with limited experimental validation or with Hobotnica, which allows for valid quality estimation of DM analysis performance for a newly generated dataset in absence of gold standard data. Evaluation of newly developed methods for DM analysis should not be performed only on simulated data due to a significant bias in the results. This bias should be at least partially compensated with tests on 'ground truth' data, no matter how limited, as well as experimental datasets, for evaluating which Hobotnica provides means.

Point 5: Additionally, the limitations and potential biases of the study are not adequately addressed.

Response 5: To improve our analysis of the possible limitations of the proposed approach, we have expanded the corresponding paragraph in the “Discussion section”

line 219:

“Detecting and assessing an approach's limitations is as important as its design or validation. In this work, we limited Hobotnica applicability to DMC only, while the DMR format for methylation signatures is often more widely used and more easily interpreted. DMR format, however, poses severe problems for signatures comparison, since setting a formal criteria, whether two intersecting yet differing regions should be treated as the same entity or not, is a challenging task. In addition, each region can be characterized not only by the mean level of methylation but also by the length of the region, as well as the density of sites in it.”

Point 6: The manuscript contains several grammatical errors, awkward sentence structures, and unclear phrases, which make it difficult for readers to understand the content.

Response 6: We thank the reviewer for bringing this problem to our attention. We have re-addressed our text and corrected minor errors and re-written sentences that needed to be rephrased more clearly.

Reviewer 3 Report

The manuscript I review contains all the chapters that original articles must contain. The manuscript was well planned. The objectives of the work are clearly defined. The methodology is properly described. The research results are extensively presented. The discussion is adequate to the research performed and the available scientific literature.
However, the manuscript must be revised before it is published. Namely, 'Introduction' is too long and needs to be shortened by at least 50%.

Author Response

Point 1: However, the manuscript must be revised before it is published. Namely, 'Introduction' is too long and needs to be shortened by at least 50%.

Response 1: We thank the reviewer for this suggestion. To address this issue, we have significantly shortened the “Introduction” section.

Round 2

Reviewer 2 Report

The authors has modified and addressed most of my concerns.

I suggest weakly accept this form of manuscript.